# Involvement of JNK1 in Neuronal Polarization During Brain Development

**DOI:** 10.3390/cells9081897

**Published:** 2020-08-13

**Authors:** Rubén Darío Castro-Torres, Oriol Busquets, Antoni Parcerisas, Ester Verdaguer, Jordi Olloquequi, Miren Etchetto, Carlos Beas-Zarate, Jaume Folch, Antoni Camins, Carme Auladell

**Affiliations:** 1Department of Cell and Molecular Biology, Laboratory of Biology of Neurotransmission, C.U.C.B.A., Universidad de Guadalajara, 45200 Jalisco, Mexico; rubendario230@gmail.com; 2Department of Pharmacology, Toxicology and Therapeutic Chemistry, Pharmacy and Food Sciences Faculty, Universitat de Barcelona, 08028 Barcelona, Spain; oriolbusquets92@gmail.com (O.B.); mirenettcheto@ub.edu (M.E.); camins@ub.edu (A.C.); 3Department of Biochemistry and Biotechnology, Medicine and Health Sciences Faculty, Universitat Rovira i Virgil, 43201 Reus, Spain; jaume.folch@urv.cat; 4Centre for Biomedical Research of Neurodegenerative Diseases (CIBERNED), Instituto de Salud Carlos III, 28029 Madrid, Spain; t.parcerisas@gmail.com (A.P.); everdaguer@ub.edu (E.V.); 5Institut de Neurociències, Universitat de Barcelona, 08035 Barcelona, Spain; 6Department of Cell Biology, Physiology and Immunology, Biology Faculty, Universitat de Barcelona, 08028 Barcelona, Spain; 7Laboratory of Cellular and Molecular Pathology, Biomedical Sciences Institute, Health Sciences Faculty, Universidad Autónoma de Chile, 3460000 Talca, Chile; jordiog82@gmail.com; 8Department of Cell and Molecular Biology, Laboratory of Neural Regeneration, C.U.C.B.A., Universidad de Guadalajara, 44340 Jalisco, Mexico; carlos.beas@academicos.udg.mx

**Keywords:** JNK1, migration, asymmetric division, synaptogenesis, NMDAR, WDR62, SCG10, MAP1B, DCX

## Abstract

The c-Jun N-terminal Kinases (JNKs) are a group of regulatory elements responsible for the control of a wide array of functions within the cell. In the central nervous system (CNS), JNKs are involved in neuronal polarization, starting from the cell division of neural stem cells and ending with their final positioning when migrating and maturing. This review will focus mostly on isoform JNK1, the foremost contributor of total JNK activity in the CNS. Throughout the text, research from multiple groups will be summarized and discussed in order to describe the involvement of the JNKs in the different steps of neuronal polarization. The data presented support the idea that isoform JNK1 is highly relevant to the regulation of many of the processes that occur in neuronal development in the CNS.

## 1. Introduction

Proper neuronal development is essential to the correct functioning of brain networks and connections. Due to its biological relevance, brain development is a controlled process that includes multiple regulatory pathways and mechanisms.

c-Jun NH2-terminal Kinases (JNKs), also known as Stress-Activated Protein Kinases, are a group of stimuli-response enzyme members of the Mitogen-Activated Protein Kinase (MAPK) family [1,2]. In mammals, JNK isoforms are codified in three different genes: Jnk1/*Mapk8*, Jnk2/*Mapk9*, and Jnk3/*Mapk10*, which express four splice variants for JNK1 (α1, α2, β1, β2), four variants for JNK2 (α1, α2, β1, β2), and ~8 variants for JNK3 (less characterized), all having a molecular weight of either 46 (p46, α1 and β2) or 54 KDa (p54, α2 and β2) with a C-terminal extension; additionally, some splice variants of JNK3 have an N-terminal extension [3,4,5]. Nonetheless, isoforms exhibit specific patterns of distribution in the organism, e.g., JNK1 and JNK2 are ubiquitous while JNK3 is only expressed in the heart, testicles and brain. Some organelles (such as mitochondria and endoplasmic reticulum) and structures (such as dendritic spines and axon terminals) show special enrichment of JNKs, usually in an isoform-specific manner. These data suggest that every isoform has different regulatory functions, vanquishing the notion of redundancy [3,4].

Different JNKs phosphorylate a wide arrange of nuclear and cytosolic substrates in serine/threonine residues, followed by proline residue, thereby underpinning their role in critical physiological processes that include neuronal functions and immunological responses, together with embryonic and postnatal development [4,5,6]. This is evidence of the highly differentiated regulatory functions of JNK isoforms [7].

In this review, we will discuss reports focused on isoform JNK1. Evidence has identified it as an essential protein in the orchestrated process of global brain development. Notably, no developmental illness has been reported to be caused by genetic mutations of the JNK1 isoform; instead, there is a large body of associative evidence. Surprisingly, this is not the case for its cytosolic substrates, such as MAP1B, MAP2, WRD62, DCX and SCG10, which, in case of impairment, can trigger neurodevelopmental disorders such as microcephaly, lissencephaly, band heterotopia, cortical dysplasia and intellectual disabilities [8,9,10,11,12,13].

### 1.1. The JNKs Signaling Cascade

In the central nervous system (CNS), JNK signaling regulates many cellular processes, ranging from morphogenesis, neuronal pathfinding, axodendritic architecture, neuronal survival, synaptic plasticity and memory formation, as well as energetic and hormonal regulation [11,13,14,15,16]. 

JNKs are prodirected serine/threonine kinases; their activation occurs in a three-step cascade of kinases, starting with the Kinase Kinase (MAPKKKs). MAPKKKs phosphorylate and activate members of the MAPK Kinase family (MAPKK), such as MAPK/ERK Kinase 4 (MKK4) and MAPK/ERK Kinase 7 (MKK7) which, in turn, activate MAPKs (JNK1/SAPK1, JNK2/SAPK2, and JNK3/SAPK3) in conserved dual phosphorylation sites of threonine and tyrosine (TPY motif sequence). Scaffold JNK-interacting proteins (JIP) facilitate the sequential phosphorylation cascade and MAPK phosphatases (MKPs) deactivate this pathway [5,17,18,19]. 

The protein structure of JNK has two globular domains which are characterized by a short N-terminal and large C-terminal peptide chain connected by a flexible domain, i.e., the catalytic site present in a deep cleft domain on the interface of lobules. Their mechanism of the action comprises multiple steps, starting with their acquisition of an open conformation in the absence of ATP, and a closed conformation when activated and bound with the substrate [20]. Several motifs are necessary to regulate the enzymatic reaction: the activation domain (A-loop) contains the TPY sequence, and a conserved G-loop (Glycine-Rich) motif is necessary to close the ATP-binding domain. The HRD motif is involved in the reaction mechanics; meanwhile, the DRG sequence is critical for conformational changing in the reaction [20,21]. JNKs also contain D-recruiting site (DRS). This sequence is necessary for interactions with their cognate sequence D-motif, found in substrates or scaffold-protein-like JIP [21]. In general, the catalytic site is conserved (See protein alignment for the main JNK isoform, Appendix A) 

Disruptions of physiological signaling of the JNKs may lead to adverse outcomes, both when the pathway is down- or up- regulated. For example, there is evidence that links dysregulations in the signaling of the JNK pathway with neurodegenerative diseases including Parkinson’s [22], Alzheimer’s [23] and Huntington’s diseases [24,25], polyglutamine diseases [26] and auditory hair cell degeneration [27]. Moreover, defects in JNK activation, with a key role in axonal growth, explain the neurodevelopmental deficits that appear in lysosomal storage diseases [1,3,4,28]. In line with this, genetic studies in patients have associated truncated mutation of JNK3, induced by translocation in chromosome 4 and Y, with severe developmental epilepsy and intellectual disability. This mutation prompts the of loss-of-function of the JNK signaling pathway, preventing the phosphorylation of the classical JNK target c-Jun and other substrates such as PSD95 [29,30]. Moreover, other studies have shown that changes in physiological signaling of the MKK7/JNK cascade may underlie the neurochemical alterations which are associated with schizophrenic symptoms [31]. Genetic association studies have linked schizophrenia with the microduplication of several genes, including TAO2 kinase, an upstream activator of the JNKs [32]. Finally, mice knockout of *Jnk1*, or mice treated with an engineered retrovirus that exclusively inhibited JNK1 in adult-born granule cells, displayed an increase in immature neurons in the dentate gyrus of the hippocampus, an effect that has been correlated with a reduction in depressive-like behaviors [33]. Our results provide evidence that a lack of *Jnk1* in mice offers neuroprotection against excitability and the cognitive impairments induced with high fat diets (HFD) [34].

### 1.2. Cytosolic Substrates of JNK in Developing Neurons

Over the years, biochemical studies have allowed us to identify multiple substrates to JNKs located in the cytoplasm or nucleus. In the nucleus, JNKs regulate multiple transcription factors which are responsible for gene expression, e.g., c-Jun, activating transcription factor 2 (ATF2), E26 transformation-specific-like 1 (Elk-1), p53, nuclear factor of activated t-cells 4 (NFAT4) and other chromatin modifiers [3]. However, bioinformatic studies of D-motifs have revealed that multiple JNK substrates may comprise cytoskeletal proteins such as actin-binding proteins, microtubule-binding proteins (MAPs), motors proteins, centrosomal proteins, basal bodies and others involved in primary cilium biogenesis [5]. Functional analyses have confirmed that JNKs phosphorylate several MAPs, including Microtubule-Associated Protein 1B (MAP1B), Microtubule-Associated Protein 2 (MAP2), Microtubule Associated Protein Tau [35], Doublecortin (DCX) [36] and superior cervical ganglion protein 10 (SCG10; also known as Stathmin 2 (STMN2) [37]. In fact, some estimates link JNK signaling with at least 60% of the total phosphorylation occurring in the growth cone during brain development, affecting substrates like the Growth Associated Protein 43 (GAP43), MAP1B, SCG10, RUN FYVE Domain Containing 3 (RUFY3) and Roundabout Guidance Receptor 2 (ROBO2) [38].

Furthermore, single neurites accumulate mRNAs of elements belonging to the JNK pathway, such as *Jnk1, Mkk4, Mkk7*, *Dual Leucine Zipper Kinase (Dlk),* a novel *SCG10*-like protein (*Sclip*), *Map2, Map1b*, and Scaffold JNK-interacting protein family members, i.e., *JipP4/JlpP* and *Jip3/JSap* [39]. Local translation rates suggest a high turnover of JNK signaling components in neurites.

### 1.3. JNKs Have Specific Functions. JNK1 is Essential for Proper CNS Development

The different JNKs isoforms have specific functions and respond differently, according to the type of stimulus, duration, cell or subcellular compartment [6,40]. In the CNS, JNK1 activity is the foremost contributor to total JNK activity; it takes part in many processes, including in the maintenance of axonal tracks during development or the control of cognitive function [4,28,41].

The use of knockout (KO) mice for the JNK1 isoform has given rise to reports that this isoform is essential for proper CNS development. It was described that the absence of a single isoform is not lethal in transgenic knockout mice, but that double null mutant embryos for JNK1 and JNK2 (*Jnk1^−/−^Jnk2^−/−^*) die between embryonic day E11 and 12. These mice fail to close the neural tube due to a decrease in apoptosis rate in the hindbrain neuroepithelium and an increase in the forebrain correlated with widespread caspase activation. Furthermore, *Jnk1^-/-^Jnk2^-/+^* mice present retinal coloboma (failure in fissure closure) and show severe defects in the lens at E18. These mice die after birth due to severe deficiencies in their development. These JNK isoforms play a different role in apoptosis, depending on the brain region where they are located, suggesting that JNK1 and JNK2 have both proapoptotic and apoptotic functions. Moreover, the absence of lethality in *Jnk1^−/−^/Jnk3^−/−^* or *Jnk2^−/−^/Jnk3^−/−^* embryos, together with the lack of apoptotic alterations in these double mutant mice, suggests that JNK1 and JNK2, but not JNK3, have a redundant role in the regulation of apoptosis in specific brain areas during brain embryogenesis [42,43]. 

*Jnk1^-/-^Jnk2^-/+^* mice present retinal coloboma (failure in fissure closure) and show severe defects in the lens at E18. These mice die after birth due to severe deficiencies in their development. Importantly, while a partial expression of JNK2 is not enough to compensate for a lack of JNK1, monoallelic expression of *Jnk1* in *Jnk1^-/+^Jnk2^-/-^* knockout mice leads to a nonobvious altered phenotype which allows the animals to survive [44]. Nonetheless, mice null for the *Jnk1* gene have the anterior commissure disrupted and exhibit alterations in the organization of cortical area, with no clear consensus regarding the consequences of this. Chang et al. reported that these mice showed growing degeneration of axonal and dendritic processes (visible in 8-month-old animals), jointly with progressive learning impairment and motor defects [41,45,46]. Our research group observed no behavioral alterations when testing 9-month-old *Jnk1^-/-^* animals using the novel object recognition test [34].

### 1.4. JNKs Regulate Programmed Cell Death During Brain Development

Neuronal death is a highly conserved and relevant process in brain development that is controlled by JNKs. It plays a role in the control of synaptic connections and morphogenesis, e.g., in the elimination of defective and abnormal cells [47]. As Kuan et al. reported, the JNK signaling pathway controls caspase activity in brain embryonic regions [43]. However, the mechanism of JNK activity in programmed cell death during development remains under study. In this line, our results showed a different number of immature neurons in the subgranular zone of the dentate gyrus of the adult hippocampus in knockouts (KOs) for JNK compared with wild type (WT) [14]. However, the mechanism of JNK activity in programmed cell death during development remains unclear. In this line, our results showed a different number of immature neurons in the subgranular zone of the dentate gyrus of the adult hippocampus in knockouts (KOs) for JNK, compared with wild type (WT) [47]. Since AGO is highly conserved across both the animal and plant kingdom, the role of the AGO1-TAK1-JNK pathway in programmed cell death in brain in mammalian development must be analyzed. In addition, it would be useful to identify the role of canonical Wnt pathway signaling in programmed cell death in the CNS in mammals, because Wnt signaling, in collaboration with the JNK pathway, promotes dorsal closure and ventral patterning during *Drosophila* embryogenesis [48]. 

## 2. Polarization of Cell Division: JNK1 and Asymmetric Division of Neural Stem Cells

The polarization process in neurons is necessary during all stages of their life. At the beginning, Neuronal Stem Cells (NCS) experience a polarized division to differentiate, and immature neurons polarize their membrane projections to migrate, the last step of polarization occurs during axodendritic biogenesis. These specific processes are forms of polarization regulated by JNKs.

The polarization of cell division is a requirement for the correct expansion and differentiation of NSCs. Around E10, radial glial cells (cortical NSC) actively proliferate in the lateral ventricles. These neural progenitors attach their plasma membrane in two subdomains: the apical membrane facing the ventricular space and the basal facing the pia (Figure 1). The apical and basal membranes of these cells represent less than 3% of the total contact; basolateral space confines the rest. The cells that experience symmetric division preserve their capacity for self-renewal to maintain cellular population (Figure 1A). However, during E12 and E15, NSCs may also start to divide asymmetrically, losing the apicobasal contacts [49].

The activity of apical polarity molecules determines the plane of excision regulated by the orientation of the mitotic spindle. These molecules are the Par protein cortical complex, which comprises aPKC (atypical Protein Kinase C), the conserved PDZ domain proteins Bazooka (Par3) and Par6 (Par3/Par6/aPKC complex), and the Gαi-LGN-NuMa centrosomic complex. The mammalian homolog of Drosophila Pins (LGN) functions as a conformational switch that links Gαi and the nuclear mitotic apparatus (NuMA) protein. LGN and Gαi may exert forces on mitotic spindles during mammalian cell division (Figure 1A,B) [50,51].

JNK1 is part of the centrosome, and reaches its maximum activity between the S-phase and the late anaphase. Hence, JNK1 regulates cell cycle mechanisms in mitotic cells [52]. JNK1 colocalizes with WDR62, a centrosomic protein with binding sites to upstream kinases such as MKK3/MKK7 and JNK1 [53,54]. Over 30 mutations in WDR62 can cause forms of primary microcephaly, often accompanied by agyria and cortical dysplasia [55].

The depletion of JNK1 and WDR62 induces abnormal spindle formation and aberrant centrosome biogenesis in NSCs during corticogenesis (Figure 1C). Knockdown of *Wr62* induced by the siRNA electroporation in utero in embryo mice cortices impairs the cell division of NSCs. As a result, NSCs reduce their population until exhaustion. The loss of self-renewal produces premature differentiation and early departure of migratory neurons, observed with supernumerary accumulation in the intermediate zone [8]. These defects resemble those seen in the developing cortices of *Jnk1^-/-^* mice [56]. Importantly, an active constituent form of JNK1 rescues the phenotype [8,57].

Moreover, we observed in our research on adult hippocampal neurogenesis that *Jnk1^-/-^* mice exhibited notable reductions of radial-glia like cells (hippocampal NSC). As a consequence, a shrinking of the ventricle layers and an accumulation of neuroblasts was observed in the intermediate zone of granule cell layers in the dentate gyrus [14]. These results agreed with observations made by Xu and colleagues [8]. In conclusion, it seems evident that JNK1 is relevant for asymmetrical division, and therefore, that it regulates the proliferation, differentiation and migration of neurons in the cortex and hippocampus. Nonetheless, diverse questions remain open in this field.

## 3. Neuronal Polarization

Neuronal membrane projections form dendrites and axons as part of a cell polarization program that takes place after the birth of the neuron. Accumulated evidence shows that neural symmetry breakage comprises two elements: an autonomous-independent cellular intrinsic mechanism and extrinsic microenvironmental effectors (polarizing cues) [58,59,60]. 

Gary Banker and coworkers initially described the intrinsic mechanism using hippocampal postmitotic neurons obtained from the brain of rat embryos at E18 [61]. They observed in their cultures that neurons, successfully seeded, extend lamellipodia over the plate, for ~12 hours (Stage 1). Then, sprouting of 4 to 5 neurites (Stage 2, or multipolar stage) arises. Subsequently, only a single “neurite” experienced growth (Stage 3). From then on, it was determined that the longest neurite was the nascent axon (axonogenesis). After axon elongation (7 days of culture), dendritogenesis occurred (Stage 4), comprising the branching of the remaining neurites. Finally, spinogenesis and axonal branching occurred in stage 5 (Figure 2) [62,63]. Now, research in this field is focused on disentangling the mechanics of this process in vivo. For example, dendritogenesis is part of a developmental process called synaptogenesis (Section 5), in which spine dendrites make contact with the axonal terminals.

### 3.1. Molecular Signalling Along Different Stages

A global-to-local activation–inhibition model explains the stereotypical behavior of neurons when they break their cell symmetry. Positive feedback signaling is primed on a specific neurite. This signal is propagated to the soma, inducing the stabilization and elongation of microtubules, and consequently, favoring the differentiation of the axon (Stage 3). A negative feedback signal or self-inhibition of the remaining neurites represses the formation of a second axon (Figure 2B,C) [59,64].

The nature of the cell signaling that precedes the sprouting of neurites remains unclear, as does the involvement of JNKs in it. Nonetheless, the positioning of the centrosome and Golgi apparatuses, and cell adhesion signaling induced by N-cadherin, play a pivotal role in determining cell symmetry breakage, just like in axon determination [65,66,67] (Figure 2A). Regulatory activities of JNKs occur mainly from stage 2 onwards.

### 3.2. Axon Determination

The involvement of JNK signaling in axon determination derives from early observations in studies in PC12 and neuroblastoma cell lines [68,69,70]. Exposing growth factors to these cells induced persistent activation of JNKs, resulting in their differentiation into neurons [68,69,70]. Also, pharmacological JNK inhibition (SP600125) and genetic tools (constitutive active *Jnk*) supported this notion. Subsequent experiments were extended to primary dorsal root ganglia explants [71] and dopaminergic neurons [72], consolidating a formal theory on the involvement of the JNKs in axon determination.

Relevantly, none of these experiments identified the involvement of a specific JNK isoform. The following section describes specific cell signaling mechanisms that respond to growth factors, and how they activate the JNKs.

#### 3.2.1. Ras Homologous (Rho) GTPase Family

Several molecules take part in axon determination, like the Transforming Growth Factor-β (TGF-β), Semaphorin3A (Sema3A), Insulin Growth factor-1 (IGF-1), brain-derived neurotrophic factor (BDNF) and others such as Wingless and Int-1 (Wnt) proteins [73]. 

Most of these molecules act through the Rho GTPases family (Rac1, Cdc42, and RhoA), which activates the Par3/Par6/aPKC complex [74]; this is essential for neural polarization and axon specification in the animal kingdom [73,74,75,76,77].

Cell division control protein 42 homolog (Cdc42) and Family Small GTPase 1 (Rac1) participate in the breakage of cell symmetry during neuritogenesis through the Par6/aPKC complex. Cdc42 activates Par6. Then, Par6/Par3 activates Rac1 through two guanine nucleotide exchange factors: T-Lymphoma Invasion-and Metastasis-Inducing Protein 1 (Tiam1) and Sif- And Tiam1-Like Exchange Factor (STEF). At the same time, Rac1 activates Phosphatidylinositol-4,5-Bisphosphate 3-Kinase (PI3K) and Cdc42, forming a positive feedback circuit [78] (Figure 2B). Moreover, the activation of Rac1 and Cdc42 leads to the regrowth of neurites, whereas Ras Homolog Family Member A (RhoA) activation collapses the growth cone. Persistent activation of Rac1 and Cdc42 leads to the activation of JNK, while activating Ras Homolog Family Member A (RhoA) leads to its deactivation [79] (Figure 2B). The final effectors of this pathway are components of the actin and microtubule cytoskeleton, which are essential for axon specification.

#### 3.2.2. WNT/Frizzled/Disheveled Pathway

The canonical pathway Wnt/β-catenin, and the noncanonical Wnt/planar cell polarity (PCP) pathway regulate cellular polarization. Wnt ligands bind the family receptor Frizzled, which, in turn, activates the Dishevelled (Dsv) factors and members of the Rho GTPases family by inhibiting Glycogen Synthase Kinase (GSK-3β) and activating JNK [80]. Wnt/Frizzled signaling induces microtubule stabilization.

Frizzled5 receptors accumulate in lamellipodia during stage 1, and once development reaches stage 2, the receptor reorders into the phyllopodia. During axon determination, Frizzled5 accumulates in the axon and is depleted in minor neurites. Thus, the Wnt pathway modulates the transition from stage 2 to stage 3 during neural polarization trough JNK activation [81].

### 3.3. Axonogenesis

Evidence that the JNK is indispensable for axonal growth was initially obtained in primary cultures of embryonic hippocampal neurons by Anthony Oliva and coworkers [82]. They proved that the pharmacological inhibition of JNK (SP600125) and a dominant-negative construction inhibited axon growth [82,83]. Their discussion emphasized that kinesin-1, a molecular motor protein that moves along microtubules, carried molecules for the formation of the signaling complexes needed for axonal growth (Figure 3). Indeed, it was discovered that kinesin-1 fluctuates between the neurites before axon determination and concentrates in the axonal growth cone during elongation [84,85]. Now, it has been shown that Kinesin forms a protein complex with JIP scaffolding proteins, facilitating the diffusion of relevant cargoes, including BDNF, TrkB [86] and JNK substrate SCG10 [87]. JIP1 and JIP3 recruit JNK and upstream kinases such as MKK7, DLK [88] and Rho GTPases [89] (Figure 3). Supporting this data, transgenic knockout *Jip^-/-^* and *Jip3^-/-^* mice presented aberrations and shrinking of axons in the commissural tract of the telencephalon during development. Also, the genetic deactivation of either element of this complex has severe effects in development [90,91,92,93].

In addition, *Map1b^-/-^* mice phenocopy all effects of null genetics of the *Jnk1* and *Jip1*, suggesting that the JNK1-JIP1-MAP1B pathway may play a crucial role in axonal growth. The opposite effect was observed with knockdown for *Jip4* in PC12 cells, and evidenced an increase in neurite growth and SCG10 phosphorylation, suggesting that the JNK-JIP4-SCG10 pathway may block axon growth (Figure 3).

Hirai et al., with cultured neurons from double knock-out mice *Dlk^-/-^Jnk1^-/-^*, observed an additive effect on axon shortening and an amelioration with taxol treatment, which is a chemical component that promotes the microtubule stabilization. These data supported the notion that DLK-JNK signaling affects microtubule dynamics through specific substrates. Specifically, the authors found that SCG10 influences the transition between stages 1 and 2. Meanwhile, DCX and MAP2 had a significant effect between stages 2 and 3 [91]. In vivo studies support the role of DLK and JNK, since deficiencies in DLX-MKK7-JNK1 signaling leads to defects in axon growth [41,94,95]. The role of JNK1 in these processes was reinforced with the high turnover of *Mkk7* mRNAs in the growth cone, allowing for high MAP1b phosphorylation through JNK1, and favoring microtubule bundling to increase neurite outgrowth [39].

### 3.4. Dendritic Organization

#### 3.4.1. Dendritogenesis

Dendritogenesis is the process through which short and dynamic neurites become dendrites with stable and more prolonged ramifications. Diverse signals control the dendritogenesis process. For example, SemaphorinA3 regulates dendritic morphology through its Neuropilin1 receptor and the Thousand and One Amino Acid Protein Kinase 1 and 2 (TAO1/TAO2). In fact, knock-down expression of *Tao2* reduces the number of secondary neurites and affects axonal length in neuron primary cultures from E18 mice [97]. These effects were like those observed with Bone Morphogenetic Protein (BMP7). This growth factor acts on dendritogenesis and the stability of microtubules via the JNK/MAP2 pathway [98]. The Wnt signaling pathway also regulates dendritic arborization. Hence, the Wnt7b ligand induces dendritogenesis through the Frizzled-7 receptor by downstream activation of DSV-1-Rac-JNK and Ca^2+^/calmodulin-dependent protein kinase II (CaMKII) (Figure 2D) [99].

#### 3.4.2. Dendrite Maintenance

JNK1 is constitutively active in the normal brain and regulates microtubule dynamics through MAP2 and MAP1B activity. Consequently, *Jnk1^-/-^* mice show a loss of microtubule integrity in dendrites with a progressive degeneration of long nerve fibers [41,100]. In the same line, *Jnk1^-/-^* mice display severe impairment in dendritic arborization in layers 2/3, 4 and 5 of the primary motor cortex [101] and in cerebellar neurons [102]. This data suggests that dendritic morphology may depend on JNK1 activity. Downstream substrates that may regulate dendrite morphology include SCG10, Macrophage Myristoylated Alanine-Rich C Kinase Substrate Like-1 (MARCKSL1) or MAP2 (Figure 2D) [101].

Moreover, BDNF-TrkB signaling regulates both dendritic arborization and axonal branching. BDNF induces the expression of MKP-1 (MAPK phosphatase) by deactivating JNK signaling, which results in increased microtubule stability. In embryo primary neuron cultures obtained from *Mkp1^-/-^* mice, neurons repress axon branches as a consequence of the persistent phosphorylation of several JNK substrates (Figure 2E) [103].

## 4. Neuronal Migration

Neurons originate in specific niches and migrate stereotypically to form nuclei or laminae (cortex and hippocampus). From their respective niche, NSCs differentiate to neurons and move to a specific layer through a migratory route. Some paths are relatively short (e.g., cortical glutamatergic neurons), while others have a long journey (e.g., cortical interneurons) (Figure 4A). Alterations in the migration mechanism disorganize of the cortical layers, resulting in Neuronal Migration Disorders (NMDs). Macroscopically, individuals that suffer these alterations manifest mental retardation, intractable epilepsy or other severe disabilities [104,105].

### 4.1. Tangential Migration

Cortical inhibitory interneurons emanate in the medial ganglionic eminence (MGE), caudal ganglionic eminence (CGE), and lateral ganglionic eminence (LGE) [105,106]. 

Interneurons from MGE reach the corticostriatal junction in mice at a single day of development (E12.5), invading the subpallium through the marginal zone (MZ) and the intermediate zone (IZ). As they enter the cortex, neurons migrate tangentially (orthogonal) to the subventricular zone. Then, they undergo a change direction to populate cortical layers (Figure 4B, upper panel) [106,107]. However, in *Jnk1^-/-^* mice, these migratory neurons delay their entrance to the subpallium, impairing migration of cortical interneurons (Figure 4B, bottom panel) as occurred with radial migration (Figure 4C). These observations were achieved by labelling interneurons with Green Fluorescence Protein (GFP) under control of the Distal-less Homeobox-(Dlx5/6) promoter and recording live brain slices. *Jnk1^-/-^* mice showed the most dominant effect, but double *Jnk1^-/-^Jnk2^-/+^* and tripe conditional *Jnk^flox/flox^Jnk2^-/-^Jnk3^-/-^* knock-out mice show exacerbated migratory defects at E13 [108,109]. 

Remarkably, in the late stages of development (E15), migratory defects are partially rescued by an unknown mechanism, and only cortical malformations and mispositioned interneurons are observed [108].

### 4.2. Radial Migration

During cortex expansion, glutamatergic new-born neurons from the lateral ventricles arrive at the deep and upper layers by radial migration, in which neurons grab large membrane projections from radial glial cells and climb through them towards the pia (Figure 4C, upper panel). When neurons get into the corresponding layer, they disengage and acquire final polarity [107,110,111,112]. But early after birth, progenitors lose basal contact during division and cross the subplate before tethering to the radial glia process [113]. At E18, *Jnk1-/-* mice showed an unusual accumulation of neuronal cells in the cortical and intermediate zone layers, supporting the notion that JNK1 regulates this step of development. Also, at E18 in *Jnk1-/-* mice, multipolar neurons accelerated their transition to the bipolar phase with subsequent pial-directed migration. Finally, at the E18 stage of development, the ventricular zone of *Jnk1^-/-^* mice was thinner, and layers 2, 3 and 4 were disarranged (Figure 4C, bottom panel) [56].

### 4.3. Intracellular Dynamics of Microtubules in Migratory Neurons

Tangential and radial migration follows three intracellular processes: (1) A transient cytoplasmic swelling or “cytoplasmic dilation” that extends the leading process in the direction of migration. (2) The movement of the nucleus toward the cytoplasmic swelling through a small constriction as a result of traction forces, and (3) Retraction of the cell rear process. Notably, axonogenesis occurs in the rear process during the bipolar phase of glutamatergic neuron migration, but interneurons first cease their movement, and then the axon grows. A microtubule network emerges from the centrosome and Golgi apparatus, supporting this dynamic behavior (Figure 5A) [106]. 

In cortical cultures, silencing the expression of *Jnk1* pauses nuclear translocation but does not impede cytoplasmic swelling. Additionally, when using inhibitor SP600125, similar results were observed, and the overexpression of activated JNK1 rescued the phenotype, supporting the idea that JNK1 is highly relevant in this process [105]. JNK1 might regulate substrates such as DCX and SGC10; promoting dynamic changes in the microtube network of the leading process (Figure 5A) and perinuclear cage (Figure 5B). The following evidence supports this idea: (a)DCX decorates microtubules in the perinuclear cage and the growth cone of the leading process [36].(b)JNK phosphorylation of Ser332, Thr326 and Thr336 in DCX reduces affinity to microtubules and causes delayed migration in cortical neuron cultures [36,114].(c)Phospho-DCX, phosphor-SGC10 and phospho-JNK overlap in the intermediate zone in developing cortices. *Jnk1^−/−^* exhibits reductions in phosphorylation rates at Ser73 of SGC10 in the cortical layers [115].

Indeed, a dynamic evaluation of a single migratory interneuron showed that the pharmacological and genetic inhibition of JNK (SP600125 and *Jnk^flox/flox^Jnk2^-/-^Jnk3^-/-^*, respectively) reduces the ramification rate of the leading process, delaying nuclei translocation and pausing cytoplasmic dilation. Additionally, in these cells, the centrosome is mispositioned at the rear process instead of the leading process (preprint version BioRxiv, [116]). 

## 5. Synaptogenesis

Adequate synapse function is an essential prerequisite for all neuronal processing, especially for higher cognitive functions like learning and memory. The process of synapse formation and maintenance—i.e., “synaptogenesis”—is considered the final step of neuronal polarization [117]. In this step, axonal growth cones navigate through a specific pathway until they come into contact with the appropriate target cells, and forms boutons. At this point, spines are developed in dendritic compartments. Boutons and spines are not static elements; they move actively and alter their morphology continuously, even in the adult brain, reflecting the plastic nature of synaptic connections [118]. Abnormal synaptic formation is associated with neurological disorders such as Rett syndromes and Fragile X Down [119]. As mentioned above, JNK signaling through cytoskeleton proteins regulates the microtubule and actin dynamics necessary for the correct formation of both the presynaptic and postsynaptic domains (Figure 2E and Figure 6) [58,120]. Different studies have reported that isoform JNK1 is highly present in both the pre- and post- synaptic compartments [121,122], supporting its participation in the relevant synaptogenesis steps: (a) axonal terminal and spine formation; (b) synaptic protein recruitment and structural scaffold formation; and (c) synaptic plasticity and transmission (Figure 6). 

Many reports have described how JNK1 regulates the formation of presynaptic domains (axonal buttons), through the SGC10 and MAP1B substrates that control microtubule dynamics [58,123,124,125], and the postsynaptic domains, through the MKK7 and MARCKSL1 substrates that control actin assembly [126,127,128]. JNK1 also regulates the trafficking from the cell soma to the dense core vesicles of the nerve terminals by phosphorylating synaptotagmin-4 [129]. In addition, the levels of presynaptic N-methyl-D-aspartate (pre-NMDA) receptors are regulated by JNK1-JIP interaction. These pre-NMDA receptors facilitate the release of neurotransmitters due to direct interaction between vesicles and membrane SNARE proteins (Syntaxin and Snap25) [121,130,131].

At the postsynaptic domain, JNK1 may interact with DCLKs (doublecortin-like kinases, DCLK1 and DCLK2), which are two close homologs genes of DCX [132,133]. DCLKs suppress the maturation of synapses, which is an important process for the structural plasticity of dendrites [134]. Moreover, JNK1 interacts with glutamate receptors and postsynaptic scaffold proteins, such as (NMDA) receptors, postsynaptic density protein 95 (PSD-95), calcium/calmodulin-dependent protein kinase II (CaMKII) and GluR2 of the α-amino-3-hydroxy-5-methylisoxazole-4-propionate (AMPA) receptor [135,136]. These interactions promote receptor clustering during synaptic transmission and synaptic plasticity. 

A comparison between WT and *Jnk1^–/–^* mice supported the hypothesis that JNK1 plays an important role in synaptic plasticity mechanisms, such as long-term potentiation or depression (LTP or LTD) or local translation. The induction of metabotropic glutamate receptor (mGluR) activity increases the phosphorylation of JNK1 substrates, including transcription factors p-c-Jun and p-ATF2 (activating transcription factor 2), in WT mice, but this did not occur in *Jnk1^–/–^* mice [4]. Additionally, it has been observed that JNK1 activation controls specific protein levels during synaptic plasticity through the AP-1 transcription factor. Thus, the JNK1–c-Jun–AP1 pathway is likely involved in the control of the loss and gain of synapses [137].

In the adult brain, JNK1 may be detected in the presynaptic and postsynaptic regions; it controls synaptic plasticity by changing synapse morphology through the cytoskeleton or scaffold proteins, or through protein expression [138,139]. An optogenetic study in adults showed that JNK1 controls spine motility, retraction and elimination in response to external inputs like stress [140]. In this line, a reduction in spine density in the apical and basal dendrites of cortical neurons was observed in *Jnk1^-/-^* mice [122]. Finally, a study with *Jip1^-/-^* mice showed enhanced memory in hippocampus-dependent tasks, indicating that JIP1-mediated JNK activation constrains spatial memory [138]. All these data reinforce the role of JNK1 in maintaining the synaptic plasticity of adult brain [138,139]. This is a critical component of the neural mechanisms underlying certain types of learning and memory [141]. More concretely, studies with mice reported that when higher levels of hippocampus-dependent learning occur, JNK1 signaling is upregulated [142]. Moreover, the lack of JNK1 alters the dendritic arborization in neurons; these types of alterations are seen in psychiatric disorders [101], supporting the role of JNK1 in the maintenance and formation of synapses, and consequently, in the correct activity of neural circuits.

## 6. Conclusions

The evidence gathered in this revision has allowed us to conclude that JNKs, and especially JNK1, do play a very significant role in the regulation of the development of neuronal structures during brain development. It seems that JNK1 signaling is responsible for the control of almost all steps of cell development, starting with the cellular division of NSCs until the final step in which neurons are established in a specific layer, forming the synaptic neuronal network. Consequently, negative modulation of JNK1 signaling activity leads to significant alterations, e.g., in the number of neurons, their morphology, neuron position and circuit connections in different brain areas. Earlier dysregulation of JNK1 and its cytosolic substrates could have even more detrimental effects in neurodevelopment. Despite having been proven to be effective in the treatment of affective disorders and cognitive loss, inhibition of JNK1 may be not suitable as a therapeutic agent in all pathological conditions. JNK substrates, instead of JNK1 isoforms, could be considered possible therapeutic agents, thereby avoiding affect JNK1 functions. More studies are needed to better understand the role of JNK signaling in mental disorders, as both the gain or loss of function of JNK signaling may contribute to various pathologies. However, restoring JNK activity in order to prevent neurodevelopmental pathologies is an attractive hypothesis.

## Figures and Tables

**Figure 1 cells-09-01897-f001:**
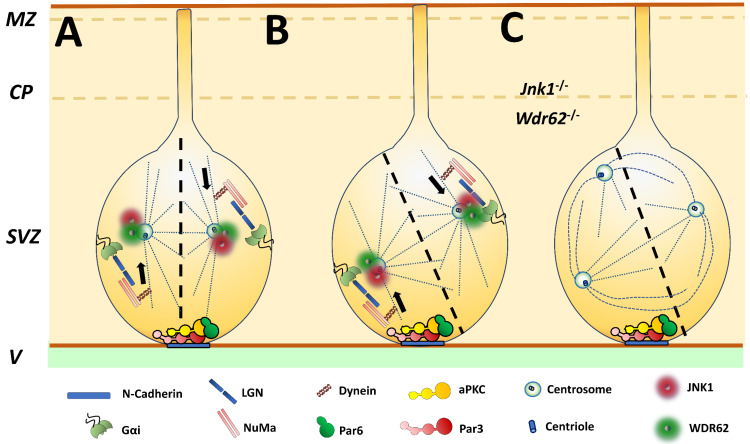
JNK1 controls polarization of dividing cells. Radial-glial-like cells (cortical NSCs) are located in the SVZ and show apicobasal polarization. The orientation of the mitotic spindle controls symmetric and asymmetric divisions: (**A**) Symmetric division preserves self-renewal and expands radial glial-like progeny, (**B**) Asymmetric division promotes the loss of apicobasal contact of daughter cells, promoting their differentiation. The activity of the Par3/Par6/aPKC and Gαi-LGN-NuMa complexes determines the orientation of the mitotic spindle. (**C**) The pericentriolar material of centrosome contains both JNK1 and WDR62. Genetic disabling of this elements induces abnormal spindle formation and aberrant centrosome biogenesis. Adapted from [50].

**Figure 2 cells-09-01897-f002:**
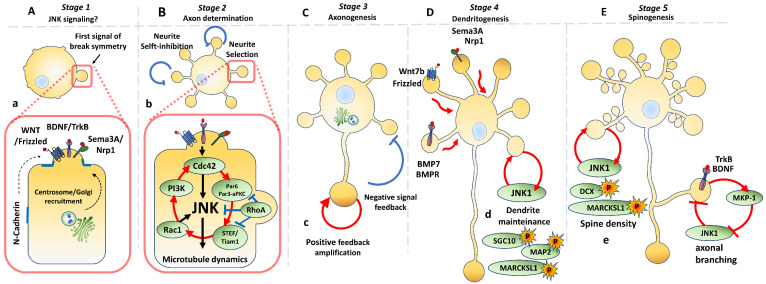
Role of JNKs in neuronal polarization. **A**–**E**: Phenotypic changes along neuronal polarization in vitro. (**A**) (*Stage 1)* Embryonic hippocampal neurons extend the lamellipodia. Twelve hours later, the breakage of neuron symmetry occurs. (**B)**
*(Stage 2)* Neurons become multipolar. One neurite is selected and experiences a positive feedback signal on its tip, which induces axon determination. (**C**) *(Stage 3, axonogenesis)* Signal propagates to the soma (red), inducing the stabilization and elongation of microtubules. The remaining neurites receive inhibitory signals (blue) (**D**) (*Stage 4, Dendritogenesis*) After axon elongation, the remaining neurites are extended. (**E**) (*Stage 5, Spinogenesis*) Finally, spines and axonal branching occur. (a–e) Molecular events that occur during neuronal polarization. (a) Axon determination is induced by growth factors. Centrosome and Golgi apparatuses are recruited. (b) Rho GTPases (Cdc42, Rac1) and the Par3/Par6/aPKC complex form a positive feedback circuit required for axon determination. (c) The persistent activation of Rac1 and Cdc42 leads to JNK activation, promoting neurite outgrowth through growth cones (red), while the activation of RhoA in minor neurites leads to JNK deactivation (blue). (d) Dendritogenesis is stimulated by BMP7 through Bone Morphogenic Protein Receptor (BMPR), Wnt7b through Frizzled receptor and Sema3A through Nrp1. Dendrite homeostasis is controlled through SCG10, MAP and MARCKSL1 substrates that are activated by JNK1 (red) (e) Spine density is regulated by substrates such as DCX and MARCKSL1, and axonal branching is promoted by the deactivation of JNK1 through BDNF.

**Figure 3 cells-09-01897-f003:**
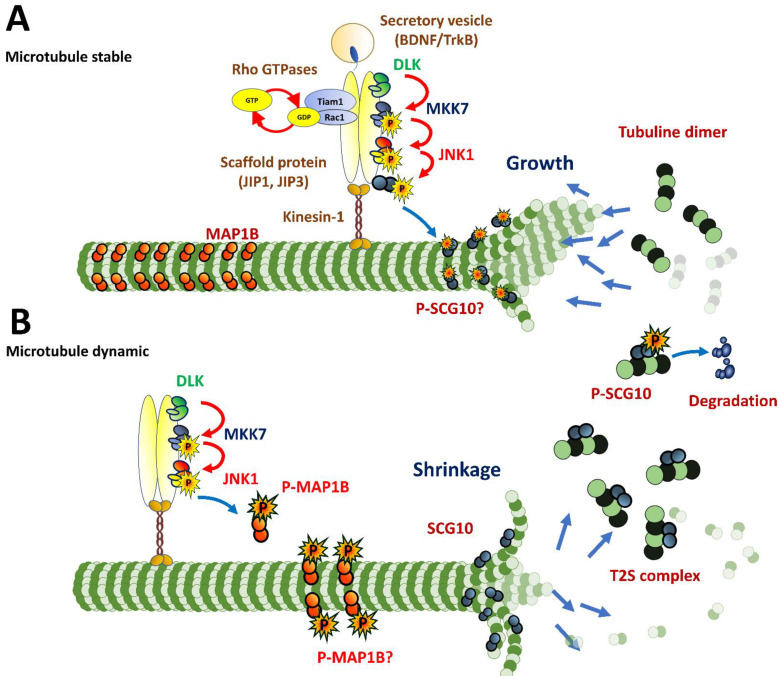
Microtubule (MT) dynamics are regulated by JNK1. (**A**) MT stability and elongation. JIP scaffolding proteins and kinesin-1 facilitate the diffusion of JNK1 and the transport of relevant cargoes needed for axonal growth such as Tiam/Rac, SCG10, BDNF and TrkB receptors. Unphosphorylated MAP1B and phosphorylated Sthatmin family members by JNK1 promote microtubule stabilization. Moreover, JNK1 phosphorylates SCG10, resulting on its degradation. (**B**) MT shrinkage. The phosphorylation of MAP1B by JNK1 and unphosphorylated SCG10 promote microtubule depolymerization. In addition, unphosphorylated members of SCG10 proteins sequestrate tubulin dimers and favor microtubule catastrophe. Adapted from [96].

**Figure 4 cells-09-01897-f004:**
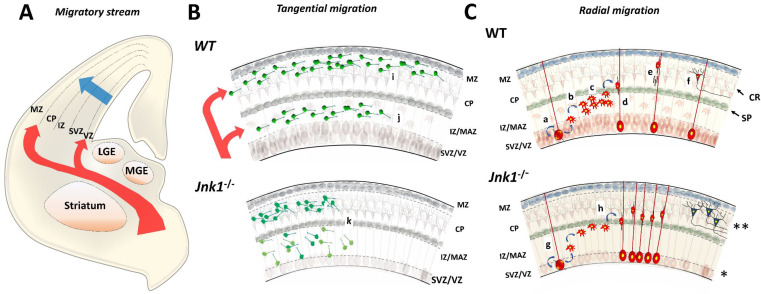
Effects of genetic Jnk1 disabling in the neuronal migration process during cortical embryonic development in mice. (**A**) Cortical glutamatergic neurons born in the lateral ventricles (blue). Cortical interneurons born in the GE (Red) (**B**) Interneurons from MGE (green) reach the corticostriatal junction at E12 and invade the subpallium. Interneurons enter to the developing cortex lateromedially following two migratory tangential streams (i, j). The (i) stream populates the superior layer, while the (j) stream populates deep layers and hippocampus. Jnk1-/- mice experience severe alterations due to a delay of entrance and erratic migration of cortical interneurons (k). (**C**) RG-like cells (red) self-renew and proliferate by symmetric division: (a-b) RG cells undergo asymmetric division and become migratory multipolar neurons. (c) Multipolar cells accumulate in IZ and form the MAZ. (d) Multipolar cells migrate towards the SP and seize radial processes to climb through them toward the pia. They acquire bipolar morphology. (e) Radial migration ceases and (f) neurons populate the upper (2/3 and 4) and deep (5 and 6) layers. This radial migration process is altered in Jnk1-/- mice at E18 of development. The multipolar-to-bipolar phase is faster than in wild-type (WT) (g to h versus a to d). The ventricular zone is smaller (*) and the cortical layers are disordered (**).

**Figure 5 cells-09-01897-f005:**
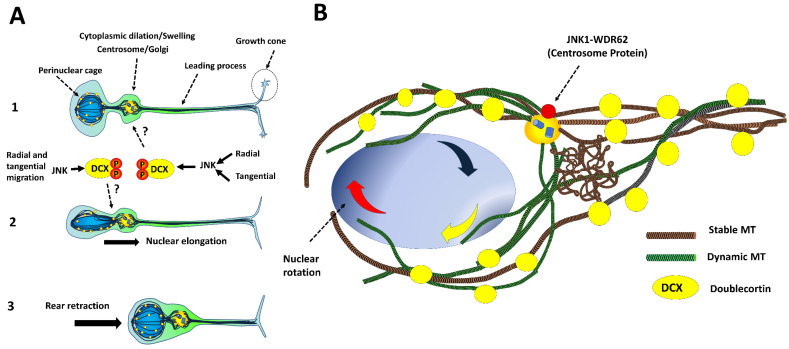
JNK1 regulates the phases of neuronal migration. (**A**) (1) Initially, the leading process is extended; (2) Later, the nucleus is pulled forward by the reorganization of microtubules that round the perinuclear cage; (3) Finally, rear retraction occurs. (**B**) Magnification of the perinuclear cage formed by MT. MTs (red and brown) are located around the nucleus (blue). The Centrosome/Golgi apparatus irradiates MTs toward the nucleus. Several targets of JNK1 are localized in the perinuclear cage (DCX, WDR62).

**Figure 6 cells-09-01897-f006:**
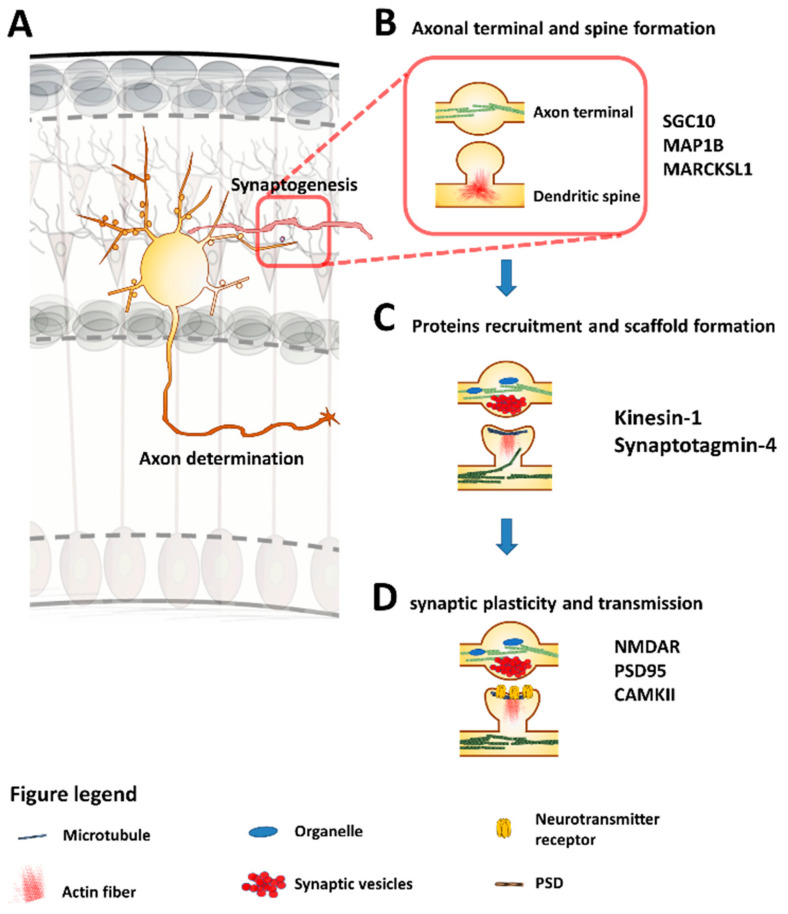
Synaptogenesis is controlled by JNK1. (**A**) Synaptogenesis occurs during neuronal differentiation. (**B**) The presynaptic (axon terminal) and postsynaptic domains (spine formation) are formed through the regulation of SGC10, MAP1B and MARCKSL1 substrates by JNK1. These events regulate actin (red) and microtubule dynamics (green). (**C**) Kinesin 1, together with JNK and JIP, control vesicle and organelle transport (blue), e.g., the formation of postsynaptic density scaffold proteins (black). (**D**) During synaptic transmission, the morphology of structures changes under the control of JNK1 (synaptic plasticity).

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
