# Peer review of "Involvement of JNK1 in Neuronal Polarization During Brain Development"

_cells, 2020, doi:10.3390/cells9081897_

Round 1

Reviewer 1 Report

The manuscript submitted by Castro-Torres et al. is well written and structured. It summarizes the current knowledge about the role of JNK1 isoform in brain development in a concise and understanding manner. The incorporation of some minor points, detailed below, would increase the interest of the manuscript for the general scientific community. Once the requested information is incorporated, I recommend the manuscript for its publication in Cells.

Minor points:

  1. It should be mentioned that JNK1 is a S/T kinase.
  2. It would be interesting to discuss the arguments that support the notion of complete redundancy of JNK isoforms. It would also be interesting to know the sequence identity between isoforms and the conservation of the catalytic domain.
  3. The cytosolic substrates that can trigger neurodevelopmental disorders should be mentioned.
  4. More information of the dysregulation of JNK1 pathway in mental disorders would be welcome. Is it a potential therapeutic target?

Author Response

Taking the comments from reviewers into account, we made the appropriate modifications in the manuscript. We thank that comments because they helped toi improve the manuscript

Response to Reviewer 1 Comments 

I attached the document

Reviewer 2 Report

The review is comprehensive and informative for understanding the role of JNK1 in neuronal development in the central nervous system. The review allowed me to understand several targets of JNK1 signaling as biomarkers or possible targets to prevent neurodevelopmental disorders.

Major comments

1. JNKs stimulate or inhibit cell death in a context-dependent manner by modulating the activities of pro- or anti-apoptotic proteins through distinct phosphorylation events. Programmed cell death is a universal feature of embryonic and postnatal neuroproliferative regions throughout the central nervous system. I would like to recommend that the authors discuss the role of JNK in programmed cell death during brain development in the revised manuscript.

2. I feel that this manuscript is too much paragraphs. I think that the author should describe in appropriate paragraphs.

Minor comments

L73; You should provide some references.

L306; Please delete one of the two '. (Period)'.

Author Response

Response to Reviewer 2

Taking the comments from reviewers into account, we made the appropriate modifications in the manuscript (in red).

We thank that comments because they helped to improve the manuscript.

I attached the documents with the responses
